# Metabolomics: A New Tool in Our Understanding of Congenital Heart Disease

**DOI:** 10.3390/children9121803

**Published:** 2022-11-24

**Authors:** Pier Paolo Bassareo, Colin J. McMahon

**Affiliations:** 1Mater Misercordiae Hospital, Mater, D07 R2WY Dublin, Ireland; 2Children’s Health Ireland at Crumlin, D12 N512 Dublin, Ireland; 3School of Medicine, University College Dublin, Belfield, D04 V1W8 Dublin, Ireland

**Keywords:** metabolomics, congenital heart disease, paediatric cardiology, mass spectrometry, nuclear magnetic resonance spectrometry, cardiac surgery

## Abstract

Although the genetic origins underpinning congenital heart disease (CHD) have been extensively studied, genes, by themselves, do not entirely predict phenotypes, which result from the complex interplay between genes and the environment. Consequently, genes merely suggest the potential occurrence of a specific phenotype, but they cannot predict what will happen in reality. This task can be revealed by metabolomics, the most promising of the “omics sciences”. Though metabolomics applied to CHD is still in its infant phase, it has already been applied to CHD prenatal diagnosis, as well as to predict outcomes after cardiac surgery. Particular metabolomic fingerprints have been identified for some of the specific CHD subtypes. The hallmarks of CHD-related pulmonary arterial hypertension have also been discovered. This review, which is presented in a narrative format, due to the heterogeneity of the selected papers, aims to provide the readers with a synopsis of the literature on metabolomics in the CHD setting.

## 1. Introduction

Congenital heart disease (CHD) represents the most common birth anomaly, with a reported prevalence approximately 8:1000 children who are born alive [1]. The genetic origin underpinning CHD has been extensively investigated over the last few decades. Undoubtedly, a great number of CHDs have their origin in genetic mutations. This is notwithstanding the fact that the same genetic background may manifest in different cardiac phenotypes, and, conversely, that similar phenotypes may arise as the result of different genetic aetiologies [2]. More recently, epigenetics has also gained increasing attention. It is defined as the study of changes in phenotype, which are caused by an environment-triggered modification of gene expression rather than alteration in the genetic code itself [3].

Genetics and epigenetics interact and lead to multiple metabolic pathways of extremely small molecules, which sometimes cross over one another. The “*omics sciences*” are very promising tools to study this complex interplay. They are defined as the disciplines in which study molecules have similar qualities. They are named “*genomics*”, which studies DNA structure and function; “*transcriptomics*”, which is focused on studying mRNA molecules; “*proteomics*”, which studies proteins (the latter are also the phenotypical manifestation of genome content); and “*metabolomics*”, the discipline which studies metabolites (small molecules, which are produced by enzymatic reactions encoded in the human genome. They interplay with each other to result in the individual metabolism). Progressing from the first one to the last, their complexity increases whilst the quantity of information decreases [4]. Genes, by themselves, do not predict exact phenotypes, as there is a complex interplay between genes and the environment. As such, genomics, transcriptomics, and proteomics merely suggest the potential occurrence of a specific phenotype, but cannot fully predict what happens in reality. This task belongs to metabolomics, the most promising of the “*omics sciences*”. Metabolomics studies metabolites and the “*metabolome*” represents the collection of all metabolites. Metabolomics provides a functional picture of an organism, as determined by the sum of his/her genes, RNA, and environmental factors [5].

This review provides an overview of metabolomics as applied specifically to the field of CHD.

## 2. Metabolomics Technique

The metabolomic analysis of the huge array of chemical compounds that possess different physical and chemical properties is carried out by two techniques, namely nuclear magnetic resonance spectroscopy and mass spectrometry. Mass spectrometry, combined with separation techniques such as gas or liquid chromatography, estimates the mass-to-charge ratio of ions in a molecule of a biological sample and represents them in the form of a spectrum (see Figure 1).

On the other hand, proton nuclear magnetic resonance spectroscopy is able to characterise low-molecular-weight metabolites containing a nuclear magnetic resonance nucleus in a biological sample and represent them in a spectrum (Figure 2).

Some of the advantages of proton nuclear magnetic resonance spectroscopy include that it is a fast, highly reproducible and non-destructive method. Furthermore, it usually does not require sample preparation and can be used for the analysis of tissue samples. However, this technique has quite a low sensitivity, may encounter trouble in quantifying co-resonant metabolites, and is relatively expensive [5].

## 3. Search Strategy

The electronic databases including Pubmed/Medline, Scopus, and Web of Science were investigated from their initial existence up to 8 October 2022. All searches were conducted using the following MeSH terms: *metabolomic(s)*, *metabonomics*, *congenital heart disease*, *congenital heart defect, paediatric cardiology*.

All studies relevant to metabolomics in the field of CHD were included in this review, with no limitation on the period of publication. However, some specific exclusion criteria were employed, including: (1) studies not focused on the selected topic (2) papers in a language other than English (3) duplicated studies and (4) those studies not available from libraries for full-text assessment.

Two hundred and twelve studies were identified through the database search. After excluding those studies that did not reach criteria mentioned above, 39 papers were included in the review.

Due to the limited number of papers, this review is in the form of a narrative review.

## 4. Prenatal Diagnosis

Although significant research has focused on the genetic and environmental factors underlying CHD, very few studies have been focused on teratogenic mechanisms.

In one study of metabolomic biomarkers in maternal first-trimester blood, mass spectrometry and nuclear magnetic-resonance-spectrometry-based metabolomic analyses were performed between 11 weeks’ and 13 weeks 6 days’ gestation. One hundred and twenty-three metabolites were detected to occur in significant quantities in those mothers with babies affected by CHD compared to normal controls. Significant differences in sphingomyelin, acylcarnithine, and other metabolite levels were detected in pregnancies affected by CHD. Predictive algorithms for CHD showed a high sensitivity (0.929) and specificity (0.932) [6].

The prenatal diagnosis of CHD typically relies on the use of foetal echocardiography, but some cases may go undetected despite this investigation. An untargeted metabolomic study performed through a combination of gas chromatography and mass spectrometry showed that amniotic fluid uric acid level had a moderate predictive power in identifying those cases with CHD. This may represent a potential diagnostic biomarker to identify pregnancies at risk of CHD [7].

When extending this analysis to the third trimester, by using the same tools to analyse urine samples, dysregulation was detected in lipid (including phospholipid biosynthesis, phosphatidylcholine biosynthesis, phosphatidylethanolamine biosynthesis, and fatty acid metabolism) and folate metabolism. This dysregulation in lipid metabolism may explain, at least in part, the elevated risk of CHD in offspring of diabetic mothers [8].

Another study evaluating the potential biomarkers of maternal urine metabolomics for foetuses with CHD, based on modified gas chromatography-mass spectrometry (during the 2nd and 3rd trimesters), revealed that the most important potential biomarkers linked to CHD were not only uric acid, but also 4-hydroxybenzeneacetic acid, 5-trimethylsilyloxy-n-valeric acid, propanedioic acid, and hydracrylic acid. These results suggest that aromatic amino acid and short-chain fatty acid metabolism may also be crucial in the development of CHD [9].

In another metabolomics study on maternal blood, the metabolomic signature of the second trimester in mothers with a baby affected by a foetal cardiac abnormality showed significant differences to that of a normal pregnancy. Malonic acid, 3-hydroxybutyric and methyl glutaric acid, urea, androstenedione, fructose, tocopherol, leucine, and putrescine were the most relevant metabolites detected between the two groups [10].

When comparing the gut microbiota and plasma metabolic profile of mothers of infants with CHD to healthy controls, differences were identified in 34 bacteria and 53 plasma metabolites. Specifically, the Bifidobacterium and Streptococcus bacteria showed positive correlations with a range of metabolites belonging to the lipid metabolism pathway. Therefore, perturbations of maternal gut microbiota and plasma metabolites may be linked to the risk of occurrence of congenital heart abnormalities in their offspring [11].

## 5. Generic Findings

On metabolomic analysis, CHD patients showed significant differences in the concentrations of a significant number of metabolites. Differences between CHD and controls were greater in number and degree than those between different CHD aetiologies. A specific group of metabolites containing amino acids and their metabolites (those of the arginine metabolic pathway including betaine, dehydroepiandrosterone, cystine, 1-methylhistidine, serotonin and bile acids) were associated with negative clinical outcomes across all anatomic CHD subgroups. [12].

A similar metabolomics study showed that the blood amino acid and choline metabolite levels in CHD patients were significantly different in comparison with healthy control children. In particular, four metabolites, betaine, taurine, glutamine, and phenylalanine, could be used as potential biomarkers in the screening and diagnosis of CHD [13].

## 6. Specific Congenital Heart Diseases

### 6.1. Bicuspid Aortic Valve

Bicuspid aortic valve is the most frequent CHD. It is responsible for valvular dysfunction and aortopathy in affected patients. Urine nuclear magnetic resonance metabolomics was employed to search for a molecular fingerprint for this particular cardiac defect. As a consequence, seven metabolites (3 hydroxybutyrate, alanine, betaine, creatine, glycine, hippurate, and taurine) were identified as potential biomarkers. Among these, glycine, hippurate and taurine individually displayed the highest sensitivity and specificity [14].

Significant differences in arginine and proline metabolisms were also identified between patients with bicuspid aortic valve and a normal trileaflet aortic valve. The observed augmented arachidonic acid metabolism may be a consequence of more severe baseline haemodynamics and worse left ventricular reserve remodelling after transcatheter aortic valve replacement in those patients with bicuspid aortic valve [15].

By using a serum-based metabolomics profiling method as a tool for bicuspid aortic valve screening, two metabolites were identified: monoglyceride and glycerophospho-N-oleoyl ethanolamine. A predictive model was established with a sensitivity of 76.7%, specificity of 90.0%, positive predictive value of 80% and negative predictive value of 85.0% [16].

### 6.2. Ventricular Septal Defect

Blood samples and thymus tissue collected from patients with ventricular septal defect underwent liquid chromatography-mass spectrometry-based metabolomics. This showed that two metabolites, uric acid and sphingomyelin, were increased in the serum of these patients [17].

### 6.3. Tetralogy of Fallot

Blood samples on case (n = 550) and control mothers (n = 221) were obtained from the National Birth Defects Prevention Study. A metabolomic profile of 22 pregnant females with Tetralogy of Fallot (TOF) and hypoplastic left heart syndrome (HLHS) was studied. Overall, there were 9 significant metabolites detected in the mothers of HLHS patients and 30 metabolites in the mothers of the TOF patients. Among them, 2 sphingolipids were significantly increased in the HLHS group, while 2 triglycerides were decreased and cholesterol esters were elevated in TOF females. These metabolites were hypothesised to possibly play a pivotal role in both cellular signalling and disease development. It was also proposed that dietary modifications may potentially prevent these two forms of CHD [18].

Metabolomic analysis performed using right atrial biopsy samples in patients with TOF discovered that carbohydrate and heme metabolism were upregulated, whereas bile acid metabolism, lipid droplet, and lipid binding were downregulated. Additionally, the butanoate metabolism-related hub regulators ALDH5A1 and EHHADH were significantly downregulated, potentially contributing to myocardial tissue damage in those patients with TOF [19].

Combining transcriptomics and metabolomics analysis, by means of liquid chromatography coupled with mass spectrometry, on right atrial specimens in TOF patients, showed a downregulation of sphingolipids. This may reflect the chronic hypoxia experienced in many patients with TOF [20].

### 6.4. Dextro-Transposition of the Great Vessels

Dextro-transposition of the great arteries is a life-threatening CHD requiring surgery, requiring the use of cardiopulmonary bypass, usually within the first month of life. Fourteen newborns with dextro-transposition of the great arteries with intact ventricular septum underwent arterial switch operation. Urine samples were collected pre- and post-surgery. Three hundred and seventy-one different metabolites were identified between the preoperative state and following surgery. Among these, 13 belonged to the kynurenine pathway of tryptophan degradation. These metabolites could be potential biomarkers of brain injury in cyanotic CHD [21].

### 6.5. Single Ventricle

In complex CHD, such as single ventricle hearts, which have been surgically palliated with the Fontan technique, abnormalities in the tricarboxylic acid cycle and amino acid metabolism were identified as distinguishing the metabolomic profiles of these patients [22].

The targeted metabolomic analysis of serum amino acids in adult Fontan patients, with a dominant left ventricle, was undertaken in one study. Blood levels of asymmetric dimethylarginine, methionine sulfoxide, glutamic acid, and trans-4-hydroxyproline and the methionine sulfoxide/methionine ratio were shown to be significantly elevated and blood levels of asparagine, histidine, taurine, and threonine were significantly reduced in the Fontan patient cohort compared to the control group.

The methionine sulfoxide/methionine ratio values had a significant negative correlation with oxygen uptake on exertion. These alterations in amino acid metabolome, identified in Fontan patients, suggest relationships between Fontan pathophysiology, altered cell energy metabolism, oxidative stress, and endothelial dysfunction similar to those found in biventricular patients with congestive heart failure [23]. In the same Fontan population, other differences detected in phospholipid and acylcarnitine metabolic pathways may explain the altered cell signalling and metabolism, chronic low-level inflammation, and alteration of functional/structural properties of lymphatic or blood vessels seen in Fontan patients. Again, all these modifications have also been reported in biventricular patients, who present with heart failure [24].

Metabolomics is not routinely used when quantifying amino acids in CHD. In one study, liquid chromatography and mass spectrometry were compared when sequencing them. Both methods proved to be reliable, although, in the Fontan patient group, concentrations significantly differed between the two techniques. Mass spectrometry is time-saving compared to liquid chromatography [25].

### 6.6. Cyanotic Congenital Heart Disease

In relation to cyanotic CHD, 12 specific metabolites were detected that differentiated between hypoxemic and non-hypoxemic patients, including lower methylmalonic acid, glutamate and hypoxanthine, and higher thymine and myo-inositol. No significant variations were observed in relation to the degree of hypoxemia between these groups [26]. One study investigated the pathophysiology of cyanotic CHD and underlying chronic hypoxia on an untargeted metabolomic analysis of myocardial tissue from 20 CHD patients with cyanosis and 15 normal controls. Overall, 71 metabolites out of the 113 that were detected in cardiac tissue differed between the two groups. Eleven amino acids were increased in cyanotic CHD individuals, demonstrating that protein synthesis was down-regulated. Furthermore, most of the metabolites in the Krebs circle (especially nicotinamide adenine dinucleotide) were increased in the cyanotic CHD subgroup, suggesting a down-regulation of aerobic energy metabolism. These analyses suggest that the aforementioned altered metabolic pathways may have been mediated by hypoxia. [27].

There is strong evidence that hypoxemia in cyanotic CHD has the potential to alter metabolic pathways. This is supported by the metabolomic mass spectrometry analysis of early postnatal amino acid and carnitine/acylcarnitine profiles in newborns with cyanotic CHD in comparison to healthy controls. The former had higher levels of alanine, phenylalanine, leucine/isoleucine, citrulline, ornithine, C5, C5-OH, but lower levels of C3, C10, C12, C14, C14:1, C16, C16.1, C18, C5-DC, C6-DC, C16-OH, C16:1-OH, when compared with the control group [28].

In cyanotic CHD, significant metabolic shifts were detected during puberty through a combination of positron emission tomography/computed tomography and magnetic resonance imaging. Heart tissue samples were collected. Prepubertal patients with cyanotic CHD had glucose-dominant cardiac metabolism and normal cardiac function. In contrast, during puberty, myocardial glucose uptake and glycolysis were significantly decreased, whereas fatty acids were significantly increased, along with decreased left ventricular ejection fraction. These clinical phenotypes were confirmed in an animal model of chronic hypoxia. HIF-1α was identified as the cornerstone of cardiac metabolic switching. It was upregulated during childhood but significantly downregulated during puberty. Chronic hypoxia hampered the switch of myocardial substrates from fatty acids to glucose, thus inhibiting ATP production and impairing ejection fraction [29].

### 6.7. Arrhythmogenic Right Ventricular Cardiomyopathy

Arrhythmogenic right ventricular cardiomyopathy (ARVC) is a potentially life-threatening inherited disease, which may cause ventricular arrhythmias and sudden cardiac death. Mutations in at least eight different genes were identified as potentially responsible for ARVC. A rare subtype of the disease is that caused by the mutation TMEM43 p.S358L in males. TMEM43 p.S358 is similar to the mutation CG8111 p.S333 found in the fly Drosophila melanogaster.

Metabolomics proved that the S333 in CG8111 is essential in lipid metabolism in the insect. Interestingly, this lipid metabolism impairment was also found in a murine model of ARVC, with fibrofatty replacement being the classic hallmark of the disease. Taken together, these findings may shed greater light on the molecular basis of the human TMEM43 p.S358L variant of ARVC [30].

## 7. Surgery Outcomes

Clinical risk factors often fail to fully explain the different outcomes after cardiac surgery. Targeted metabolomic analysis of plasma from newborns enrolled in the *Corticosteroid Therapy in Neonates Undergoing Cardiopulmonary Bypass trial* was performed to determine possible links with surgical outcomes. Blood was sampled three times, i.e., before skin incision, just after cardiopulmonary bypass, and 12 h following surgery. Single and composite outcomes were evaluated: morbidity, mortality, cardiac injury, hepatic injury, and acute kidney injury. The metabolomics profile was determined by a combination of high-resolution tandem liquid chromatography and mass spectrometry. A total of 34 metabolites changed significantly between all the measured timepoints, in terms of either an immediate postoperative decrease or increase, with a trend toward the preoperative baseline. Many of these metabolites were capable of predicting the occurrence of morbidity, mortality, cardiac, and hepatic injuries. However, the metabolomics plasma analysis failed to find any predictor of kidney injury [31].

On the other hand, acute kidney injury remains a significant cause of morbidity and mortality following CHD surgery. Progress in the scientific knowledge of this field is limited because of the poor evidence for the underlying altered metabolic pathways. As such, metabolomics may help tot identify the mechanisms of renal failure and provide potential therapeutic targets. In a piglet model of cardiopulmonary bypass with deep hypothermic circulatory arrest, targeted metabolic profiling of kidney tissue, urine, and plasma was performed to detect the specific metabolic changes in animals with acute kidney injury on histology. The tissue metabolic profile in injured kidneys proved to be different in comparison to uninjured kidneys, as evidenced by dysregulation in tryptophan and purine metabolism. On urine analysis, nine metabolites differed significantly in piglets with acute kidney injury, with a pattern suggesting increased aerobic glycolysis. No significant differences were found following plasma analysis. No overlap was found concerning dysregulated metabolites in kidney tissue and urine [32].

Not only did marked metabolomic differences exist between infants with or without acute kidney injury after cardiothoracic surgery with cardiopulmonary bypass, but a portion of patients with mild injury showed metabolic changes similar to those seen in the moderate-to-severe stage of the disease. Prominent changes to purine were confirmed, but methionine and kynurenine/nicotinamide metabolism were also involved, suggesting a potential role for metabolomic analysis in the evaluation of lower-stage injury [33].

Liquid-chromatography-tandem mass spectrometry metabolomics showed that, in infants undergoing cardiopulmonary bypass, circulating serum kynurenic acid metabolites may explain adverse neurologic and immunologic activities [34]. In an Italian study, pre-surgery urine metabolomics appeared to be able to predict long-term neurodevelopmental outcome in children with CHD. High levels of accumulation of citric-acid-cycle intermediates and glucose were found. These suggested a possible switch to anaerobic metabolism as the underlying cause of these findings [35].

Mortality for infants undergoing complex CHD surgery is over 10%, with a risk of complications up to 40%. The early identification of high-risk infants may prove challenging. Metabolomic fingerprinting of newborns ≤120 days undergoing cardiopulmonary bypass showed significant shifts in metabolic pathways, which were associated with mortality and length of cardiac intensive care unit stay. Non-survivors and individuals requiring a longer intensive care unit stay showed distinct modifications in specific metabolites, including aspartate and methylnicotinamide, which may discriminate sicker patients from those experiencing a more benign course [36].

Dysregulated metabolism occurs following surgery for CHD, contributing to organ failure and morbidity. Based on this premise, the effect of postoperative tight glycemic control compared to conventional blood glucose management on metabolism in children undergoing CHD surgery was studied. Blood plasma metabolome was profiled using proton nuclear magnetic resonance spectroscopy. Changes in metabolic profile were seen over the time course from surgery to recovery, compared with the preoperative state. Interestingly, tight glycemic control did not significantly alter the metabolic profile, while conventional glucose management did. Eight metabolites (3-D-hydroxybutyrate, acetone, acetoacetate, citrate, lactate, creatine, creatinine, and alanine) were identified as associated with surgical and disease severity [37].

## 8. Congenital Heart Disease and Pulmonary Arterial Hypertension

Unrepaired CHD with a long-standing left-to-right shunt may cause irreversible pulmonary arterial hypertension (PAH) (Eisenmenger syndrome). The initial increase in blood flow to the lungs triggers vasoconstriction, and then vascular changes due to intimal layer hyperplasia, hypertrophy of smooth cells, and thrombosis. Ultimately, due to a significant increase in pulmonary vascular resistance, the left-to-right shunt becomes a right-to-left shunt, resulting in marked hypoxemia and cyanosis and, eventually, right heart failure and death [38].

Metabolome profiling data showed that metabolites which are involved in purine, glycerophospholipid, galactose, and pyrimidine metabolisms, were significantly modified in PAH with CHD and left-to-right shunt. Alterations in microbial flora from the lung microbiota were also found. Based on these premises, pulmonary microbes and metabolites may prove to be effective biomarkers and provide valuable insights into clinical diagnosis, treatment, and prognosis of the secondary disease [39].

The molecular mechanisms underlying PAH in ventricular septal defects are unclear. A multi-omics study involving proteomics, metabolomics, and bioinformatics discovered 191 differential metabolites between patients with and without PAH. These included an increase in serotonin, taurine, creatine, sarcosine, and 2-oxobutanoate, and a decrease in vanillylmandelic acid, 3,4-dihydroxymandelate, 15-keto-prostaglandin F_2α_, fructose 6-phosphate, l-glutamine, dehydroascorbate, hydroxypyruvate, threonine, l-cystine, and 1-aminocyclopropane-1-carboxylate levels. By combining the results obtained through these different techniques, a risk score was suggested as a predictor of PAH development in patients with ventricular septal defects [40].

Nuclear magnetic resonance analysis of plasma in adult patients with CHD and associated PAH showed a significant correlation between metabolites such as lactate and threonine and mean pulmonary arterial pressure, pulmonary vascular resistance, and N-terminal pro-B-type natriuretic peptide. Lactate and threonine have the potential to become biomarkers of this disease, thus providing a non-invasive diagnosis in addition to predicting the prognosis of PAH associated with CHD [41].

Explanted lung tissue from 13 idiopathic PAH patients, 5 PAH associated with CHD patients, and 16 controls were analysed for untargeted metabolomic analysis with liquid chromatography associated with mass spectrometry. Significant differences were detected in metabolites and metabolic pathways between the PAH subgroups and control tissues. Spermine levels were significantly correlated with the patients’ cardiac output, while (2e)-2,5-dichloro-4-oxo-2-hexenedioic acid was notably correlated with serum creatinine levels. PAH subjects with higher thymine levels were found to have a better prognosis [42].

In another study, blood samples were used instead of lung specimens. Forty subjects with idiopathic PAH, 20 individuals with CHD-associated PAH, and 20 healthy controls were enrolled. Analyzing their samples through liquid chromatography along with mass spectrometry, distinct metabolite signatures distinguished idiopathic PAH and CHD PAH patients from healthy controls. Twenty-six and 15 distinct metabolites were detected, respectively, between the groups. Dysregulation involved lipid, glucose, amino acid, and phospholipid metabolisms in PAH patients in comparison with their healthy counterparts. Among these metabolites, a group from lipid metabolism and fatty acid oxidation demonstrated a close association with PAH [43].

Again, following a mass spectrometry metabolomics analysis of plasma, seventeen metabolites were identified that responded to shunt surgical or interventional closure. These correlated with clinical measurements including diastolic pulmonary arterial pressure, bicarbonate content in radial artery, bicarbonate content in superior vena cava, and the partial pressure of dissolved CO_2_ in radial artery blood samples. The metabolites and clinical parameters were, in turn, correlated with mean arterial pressure. These metabolomic patterns may be used as suitable non-invasive markers in disease monitoring [44].

## 9. Discussion

The presented scientific reports show that metabolomics has been widely used in attempts to make a prenatal diagnosis of CHD. Three maternal samples, namely urine, blood, and amniotic fluid, have been used. Many different metabolomic biomarkers have been identified depending on the metabolomic technique, the type of maternal sample, and the trimester of pregnancy. However, some overlapping results have been found concerning derangements in lipid metabolism and increased uric acid. This is consistent with previous reports concerning the decreased metabolic capacity to utilize released free fatty acids in infants with CHD, mostly in those with a cyanotic cardiac defect. Lipid metabolism seems to be most frequently disturbed in this setting [45]. Typically, fatty acids are used as a fuel to produce energy in the form of adenosine triphosphate (ATP). These fatty acids sequestered by cardiomyocytes are not only used as substrates for energy production, but, additionally, for the synthesis of triglycerides and the replacement of fatty acid chains in cell membrane phospholipids. Alterations in fatty acid metabolism affect the structure and function of the heart and are commonly seen in heart failure [46].

Hyperuricaemia, due to reduced renal excretion in concert with uric acid overproduction, is often encountered in cyanotic CHD patients, most commonly in those over 15 years of age [47]. Blood levels of uric acid are directlu related to the degree of polycythaemia, which arises secondary to chronic hypoxia [48].

When comparing cyanotic and non-cyanotic CHD, the most relevant differences are that, in the latter, protein synthesis is reduced, whilst Krebs circle is up-regulated at the expense of aerobic energy metabolism. Protein synthesis requires significant energy, which is unfortunately limited in this setting. As the oxygen supply to the heart is decreased, mitochondrial oxidative phosphorylation declines. However, lactate levels are high in cyanotic CHD patients, suggesting that anaerobic energy metabolism is up-regulated amid hypoxia.

Metabolomics also has the potential to predict outcomes after cardiac surgery. This is important, given that clinical risk factors themselves are often unable to fully explain the different outcomes after cardiac surgery. Specific metabolites are associated with morbidity/mortality, length of intensive care unit stay and cardiac, brain, hepatic, and acute kidney injuries. Renal injury by metabolomics is not detected with plasma analysis, but requires renal tissue specimens for analysis.

Peculiar metabolomic fingerprints for some specific CHD subtypes have been identified. They are very heterogeneous. At times, they are related to secondary heart or other organ failure. At other times, they are likely to be linked to the metabolic error, which is responsible for that specific CHD. Further research is required before this intriguing relationship is fully elucidated. Characteristic metabolomic patterns in repaired and unrepaired cardiac defects with PAH have also been discovered. They differ significantly from other PAH aetiologies, indicating that this is a multifaceted disease process.

In conclusion, recent advances in the “omics sciences” have revealed new opportunities in CHD research, with multiple potential applications [49]. Untargeted metabolomics is a powerful tool, which can shed light on some of the more obscure features underlying the onset of CHD. Metabolomics can represent the next step in precision cardiology, potentially differentiating between two subjects with the same genetic mutation, who manifest and do not manifest the cardiac defect. Discovering affected metabolic pathways may help to reveal the disease pathogenesis in addition to potential biomarkers for that disease. However, metabolomics specific to CHD is still in its infant phase. Therefore, to reap the benefit of this exciting new technology, and to allow for its larger clinical application, multicentre studies enrolling a large number of patients are warranted.

## Figures and Tables

**Figure 1 children-09-01803-f001:**
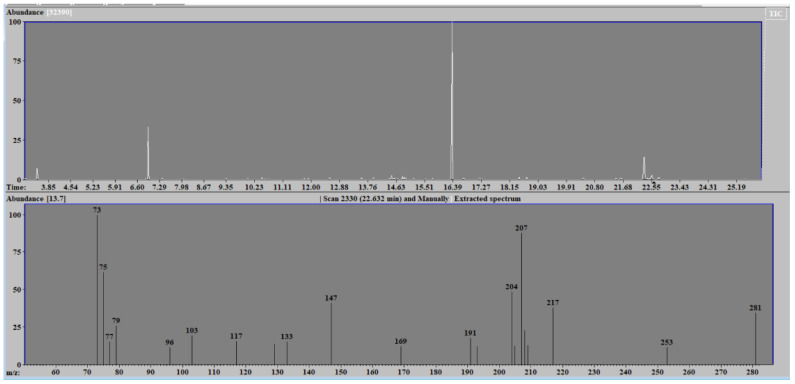
Example of mass spectrometry spectrum.

**Figure 2 children-09-01803-f002:**
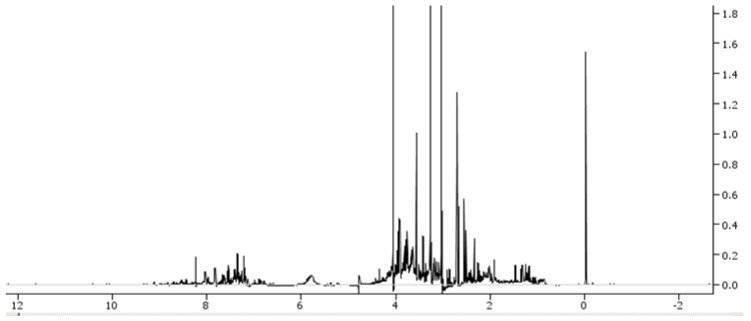
Example of nuclear magnetic resonance (NMR) spectrum.

## Data Availability

Not applicable.

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
