# Peer review of "Metabolomics: A New Tool in Our Understanding of Congenital Heart Disease"

_children, 2022, doi:10.3390/children9121803_

Round 1
Reviewer 1 Report
This well-written narrative review documents the use of metabolomics in all areas of congenital heart disease (CHD) research and medicine, including the areas of prenatal diagnosis, surgical outcomes, prediction of a variety of fetal or maternal CHD subtypes, and associations with pulmonary arterial hypertension.
Major comments:
1. Could the authors find any clearer conclusions among all these studies if comparing cyanotic and acyanotic disease?
Minor comments:
1. Caption for Figure 2 (Line 67) should be an NMR spectrum, rather than MS, correct?
2. Could authors have received publications not currently available through their library through interlibrary loan to up the number of studies included in this review?
3. Line 85, remove “s” from “hundreds”.
4. Lines 177 and 181: use the term “females” instead of “ladies”.
5. Lines 240 and 277, back end of sentence was moved to next line.
Author Response
Major comments:
- We have tried to clarify the conclusions from the 39 papers reviewed. A few sentences regarding the difference in metabolomic profile between cyanotic and acyanotic congenital heart disease patients has been added.
Minor comments:
- The caption of Figure 2 has been corrected to an NMR spectrum.
- We had difficulty accessing interlibrary publications and have limited the publications to the 39 papers in this review.
- Line 85: we have removed s from hundreds.
- We have changed ladies to females.
- The back end of the sentence is corrected.
We thank the reviewer for their helpful comments.
Reviewer 2 Report
I would you like to congratulate you for this work.
A good review on a topic that is gaining more and more interest in all fields, and also in the field of congenital heart disease.
The structure of the review is correct, treating all the topics in a structured way.
Author Response
Thank you for the time spent in reviewing our paper and your helpful comments.
Reviewer 3 Report
The work is scientifically sound and comprehensive
Author Response

(The authors gave the same response as above.)
